

# Effect of acute anaerobic performance on zinc alpha 2 glycoprotein, apelin and lipasin levels

Şaban Ünver[1], İlknur Bıyık[2], Tülin Akman[1], Emre Şimşek[3], Hamza Küçük[1], Abdurrahim Kaplan[4], Deniz Günay Derebaşı[1], Selma İşler[5], Canberk Çınar[6], Tuba Kızılet[7] and Yeliz Tanrıverdi Çaycı[2]

[1] Department of Coaching Education, Faculty of Sports Science, Ondokuz Mayis University Samsun, Samsun, Turkey
[2] Department of Medical Microbiology, Ondokuz Mayis University Samsun, Samsun, Turkey
[3] Department of Coaching Education, Faculty of Sports Science, Erciyes University, Kayseri, Turkey
[4] Department of Coaching Education, Faculty of Sports Sciences, University of Hitit, Çorum, Turkey
[5] Faculty of Medicine, Physical Medicine and Rehabilitation, Ondokuz Mayis University Samsun, Samsun, Turkey
[6] Public Health Laboratory, Ministry of Health, Sinop Provincial Health Directorate, Sinop, Turkey
[7] Department of Coaching Education, Faculty of Sport Sciences, Marmara University Istanbul, İstanbul, Turkey

Corresponding author
Şaban Ünver, saban.unver@omu.edu.tr

## ABSTRACT

The aim of this study was to investigate the effects of acute anaerobic exercise on serum levels of adipokines Zinc-α2-glycoprotein (ZAG), apelin, and lipasin. Eighteen male athletes who actively played soccer and trained at least four days a week, with a mean age of $19.11 \pm 2.59$ years, body weight of $70.61 \pm 8.17$ kg, height of $176.0 \pm 7.71$ cm, sport age of $7.22 \pm 2.60$ years and BMI of $22.76 \pm 1.68$ kg/m2 participated in the study. Athletes were subjected to the Running-Based Anaerobic Sprint Test (RAST) for anaerobic performance. Blood samples were collected from the athletes 4 times (at rest, 10 minutes, 60 minutes, and 24 hours after exercise). The results of the study showed that acute anaerobic exercise significantly increased ZAG levels ($p < 0.05$). However, no statistically significant difference was detected in apelin and lipasin levels ($p > 0.05$). In conclusion, the findings of this study indicate that acute anaerobic exercise is associated with an increase in ZAG levels, but not apelin or lipasin levels. The observations suggest that ZAG may have a specific response to anaerobic exercise, which provides valuable insight into its potential impact on energy metabolism.

# INTRODUCTION

Exercise is done both for health and to improve performance. Acute or sudden exercise leads to the formation of free radicals depending on its intensity, duration, and type. The effects of exercise on the endocrine system are related to the release or production of hormones. In this process, the release of some hormones increases whilst others decrease (*Arıkan & Akın, 2019*; *Kucuk et al., 2024*). The relationship between performance and physiological

variables has attracted researchers' attention and has been intensively studied. In this context, exercises performed at different intensities and loads trigger the secretion of different proteins in the organism. For anaerobic exercise, an excessive expenditure of force is required for a short period, irrespective of the oxygen required for the cell's energy needs.

Zinc-α2-glycoprotein (ZAG) is an adipokine encoded by the AZGP1 gene, known for its pivotal role in energy metabolism within adipose tissue (*Fan et al., 2021*). With a molecular weight of approximately 43 kDa, ZAG functions as a soluble protein with diverse physiological impacts (*Kon & Suzuki, 2022*). It acts as a lipid mobilizing factor, facilitating lipid metabolism and enhancing glucose utilization, thereby influencing insulin sensitivity (*Wei et al., 2019*). ZAG's association with metabolic syndrome, including obesity, hypertension, and dyslipidemia, underscores its broader implications in human health (*Alenad et al., 2022*).

While direct evidence linking acute anaerobic performance to ZAG remains limited, its established role in energy and lipid metabolism suggests a potential relevance to anaerobic exercise adaptations (*Kon & Suzuki, 2022*). Further exploration of ZAG's response to exercise stimuli could elucidate its dynamic involvement in metabolic processes under varying physiological demands.

Apelin, a novel bioactive peptide hormone, is presented and secreted from various tissues, including skeletal muscle (*Tatemoto et al., 1998*). Apelin receptor was first identified in 1993, followed by the isolation of the apelin molecule as the endogenous ligand of this receptor in 1998 (*Beltowski, 2006*). Apelin has been detected in endothelial cells throughout the vasculature, including arteries, veins, and small vessels in humans (*Kleinz & Davenport, 2005*) and has been implicated in cardiovascular function (*Katugampola & Davenport, 2003*), anterior pituitary function and regulation of fluid homeostasis (*Reaux et al., 2001*).

Lipasin is an enzyme that breaks down triglycerides in the blood and releases fatty acids for energy use (*Yu et al., 1999*). This enzyme plays a critical role in lipid metabolism by breaking down triglycerides carried by lipoproteins into free fatty acids and glycerol, which can be used for energy by various tissues (*Wang & Eckel, 2009*). Lipasin, also known as betatrophin, is a protein that regulates lipoprotein lipase (LPL) activities (*Karaman, Arslan & Gürsu, 2022*). Lipasin, also known as betatrophin, is also known as angiopoietin-like protein 8 (ANGPTL8) (*Lickert, 2013*). It also plays an important role in the regulation of glycolipid metabolism. In the research, it is thought that lipasin will replace insulin in the effective treatment of diabetes (*Yue et al., 2016*).

ZAG is a protein and is also known as adipokine because it is also secreted from adipose tissue. It can act in many different organs and tissues. With the assumption that ZAG, lipasin, and apelin secretions will be stimulated by acute anaerobic exercise, this study investigated the effect of acute anaerobic (high-intensity maximal exercise) performance on ZAG, apelin, and lipasin.

The results of this study will shed light on all stakeholders, including those who play sports for health, coaches, athletes, and patients. In addition, it will contribute to both sports sciences and medicine and will be a model for researchers who will work in this field. It is thought that the follow-up of these parameters in terms of monitoring both

| Experimental Design | | |
|---|---|---|
| Step 1 | Step 2 | Step 3 |
| Rest | Acute Anaerobic Performance | Post Exercise |
| Blood samples | Running –Based Anaerobic Sprint Test (RAST) 10 sec recovery → 35 m sprint → 10 sec recovery 6 repetitions | Blood samples: 10th minute; 60th minute; 24th hour |

**Figure 1  Experimental design.**

pro-inflammatory and anti-inflammatory processes in response to exercise will make significant contributions to the literature.

# MATERIALS AND METHODS

## Research design and subjects

This study was conducted using a single-center design and employed pre-test, post-test controlled experimental methods. Eighteen male professional athletes who actively play soccer and train at least four days a week, with an average age of 19.11 ± 2.59 years, a body weight of 70.61 ± 8.17 kg, a height of 176.0 ± 7.71 cm, a sports age of 7.22 ± 2.60 years and a BMI of 22.76 ± 1.68 kg/m$^2$ participated voluntarily in this study. G*power 3.1.9.2 software (Germany) was employed to determine the sample size. The sample size was determined using a medium effect size of 0.40, a statistical significance level of 0.05, and a high statistical power of 0.95, showing that 18 subjects were sufficient. The inclusion criteria for soccer players were as follows: they must not have had a serious injury (such as fractures, muscle tears and meniscus tears) within the past year, must be free from chronic injuries and diseases, must not be using chronic medications, must be aged between 17 and 23 years, must have a sports experience of at least 5 years, and must not smoke or consume alcohol. Additionally, athletes were instructed to refrain from any training or exercise beyond daily living activities for 2 days prior to the measurements and until the 24th hour following the measurements. Demographic information (age, height, body weight, sport age) of the participants was recorded. All participants were informed about the experimental procedure and the purpose of the study and their written informed consent was obtained. The study was approved by the 2020/46 ethics committee decision of the Ondokuz Mayıs University Clinical Research Ethics Committee. Moreover, this study was conducted in accordance with the ethical guidelines for human research of the Declaration of Helsinki.

## Study design

Figure 1, experimental design of the experiment conducted in this research.

### Running-Based Anaerobic Sprint Test

Anaerobic performances of the athletes were measured with the Running-Based Anaerobic Sprint Test (RAST) test. RAST, New Test-Power Timer 1.9.5. (Newtest, Oulu, Finland) brand tool was used. In the RAST protocol, the athlete performs six consecutive 35-meter sprints, taking a 10-second break between sprints. The athlete started the test by doing the first sprint. After a 10-second break, the Power Timer gives an audible signal and the athlete performs the 2nd sprint. After six sprints, the test was completed. The test offers maximum power, average power, minimum power, and Fatigue Index values (*Zagatto, Beck & Gobatto, 2009*).

### Blood collecting process and analysis

Approximately 5 ml of blood sample was taken from the arm vein by a specialist nurse to determine the athletes' ZAG, apelin, and lipasin levels. Blood collection was performed a total of four times for each athlete on the exercise day, before anaerobic performance at rest and after anaerobic performance (10 min later, 60 min later, and 24 h later). Serum was obtained from blood samples after 15 min of centrifugation (3000 rpm 4 °C). Sera obtained after centrifugation were stored at −80 °C until analysis. ZAG, apelin, and lipasin analyses were performed in the microbiology laboratory in accordance with the recommendations of the commercial kit manufacturer. ZAG, apelin, and lipasin concentrations were studied by the Enzyme-Linked Immuno Sorbent Assay (ELISA) method with a commercially available kit.

### Statistical analysis

All statistical analyses were conducted using the SPSS 21 (IBM Corp., Armonk, NY, USA). The Shapiro–Wilk test revealed that the data were not normally distributed. Descriptive statistics are presented as the arithmetic mean and standard deviation. Friedman tests were used for repeated measurements. The statistical significance was accepted as $p < 0.05$.

## RESULTS

Figure 2 shows the change in ZAG between times. Regarding ZAG values at different time points, a statistically significant difference was found between the measurement times ($p < 0.001$). There was a statistically significant difference between the ZAG value measured 10 min after exercise and the values measured at rest, 60 min, and 24 h post-exercise.

Figure 3 shows the change in apelin between times. In apelin values at different time points, no statistically significant difference was found between the measurement times ($p > 0.05$).

Figure 4 shows the change in lipasin between times. Regarding lipasin values at different time points, no statistically significant difference was found between the measurement times ($p > 0.05$).

## DISCUSSION

This study aimed to investigate the effect of acute anaerobic performance on ZAG, apelin, and lipasin concentration. The main finding of this study is that ZAG concentration

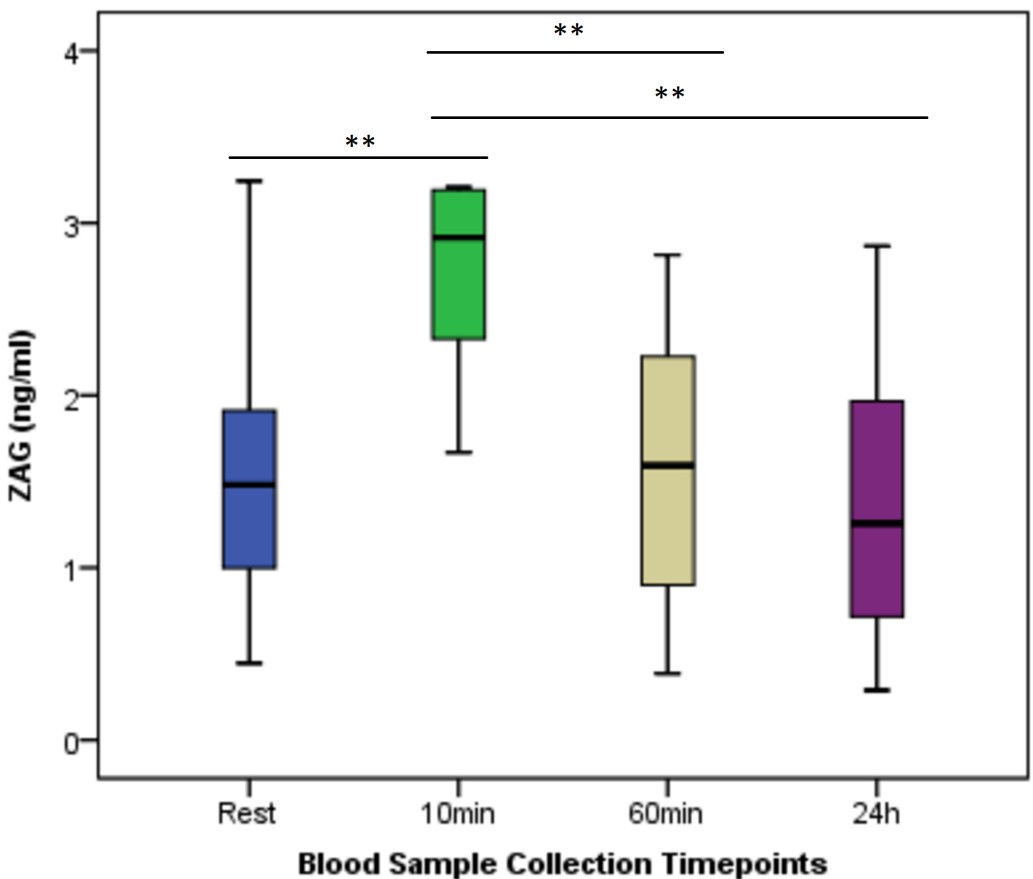

**Figure 2  Serum ZAG concentrations at rest and after anaerobic performance.** The change in ZAG between times. Regarding ZAG values at different time points, a statistically significant difference was found between the measurement times ($p < 0.001$). There was a statistically significant difference between the ZAG value measured 10 min after exercise and the values measured at rest, 60 min, and 24 h post-exercise. **$p < 0.01$; Mean ± Sd; Rest: 1.60 ± 0.79; 10 min: 2.68 ± 0.56; 60 min: 1.60 ± 0.75; 24 h: 1.37 ± 0.78; $p = 0.000$.

increased after acute anaerobic exercise. There is no change in apelin and lipasin concentrations. The data obtained suggest that acute anaerobic exercise may stimulate ZAG secretion.

To the best of our knowledge, the present study is the first to show that ZAG concentration increased with acute anaerobic exercise. Previous studies have focused on determining the effect of long-term exercise or a single resistance exercise on ZAG concentration. Accordingly, the current study is the first and original study.

In a study designed to investigate the effects of a six-month weight loss (WL) or aerobic exercise (AEX) intervention on skeletal muscle ZAG mRNA levels and protein expression, ZAG expression did not change after six months of WL or AEX (*Ge, Li & Ryan, 2023*). In another study, the effect of a single resistance exercise on ZAG concentration was investigated. Nine healthy men performed resistance exercises (bench press and leg press) with a total of five sets of 10 repetitions at 70% of maximal strength with a 90-second

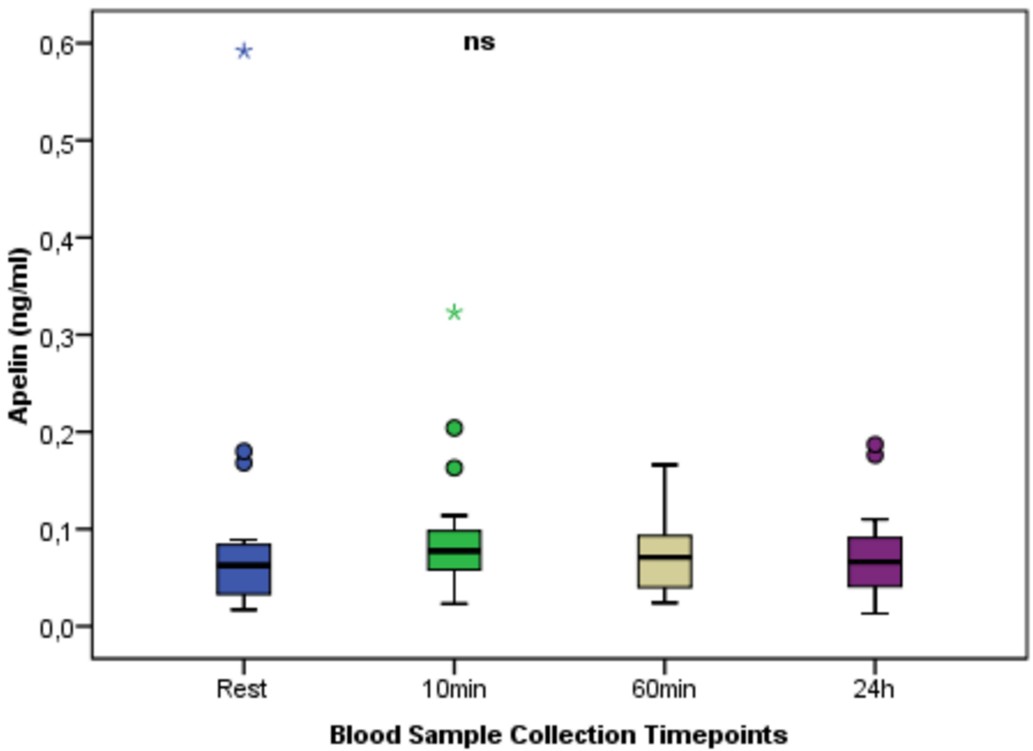

**Figure 3** **Serum Apelin concentrations at rest and after anaerobic performance.** The change in apelin between times. In apelin values at different time points, no statistically significant difference was found between the measurement times ($p > 0.05$). ns: non-significant $p > 0.05$; Mean $\pm$ Sd; Rest:0.09 $\pm$ 0.13; 10 min:0.09 $\pm$ 0.07; 60min:0.07 $\pm$ 0.04; 24 h:0.07 $\pm$ 0.05; $p = 0.559$.

rest interval between sets. Researchers suggested that serum ZAG concentration increased significantly following acute resistance exercise (*Kon & Suzuki, 2022*). In an animal study, it was reported that ZAG concentration increased following short-term intense exercise (*Cordova & Alvarez-Mon, 1995*). In the present study, it was found that ZAG concentration increased 10 min after acute anaerobic exercise and the results were consistent with previous studies. The results obtained the role of ZAG in energy and lipid metabolism and a potential link with anaerobic exercise (*Kon & Suzuki, 2022*).

Plasma apelin concentration has been well-documented to increase in individuals with obesity and type 2 diabetes (*Singla, Bardoloi & Parkash, 2010*). *Kechyn, Barnes & Howard (2015)* and *Bilski et al. (2016)* reported an increase in plasma apelin levels in response to maximal exercise in humans. Immediately after acute sprint interval exercise (*Kon, Yoshiko & Nakagaki, 2021*) and in a different study (*Kon & Suzuki, 2022*) it was reported that a single resistance exercise significantly increased apelin concentration 60 min after exercise compared to the pre-exercise value. Study in type 2 diabetic patients have shown that chronic exercise training can lead to an increase in plasma apelin levels. After 12 weeks of aerobic training, a significant increase in plasma apelin levels was observed in obese patients with type 2 diabetes (*Kadoglou et al., 2012*). In another study, aerobic (not resistance) exercise induced a significant increase in plasma apelin levels (*Kadoglou et*

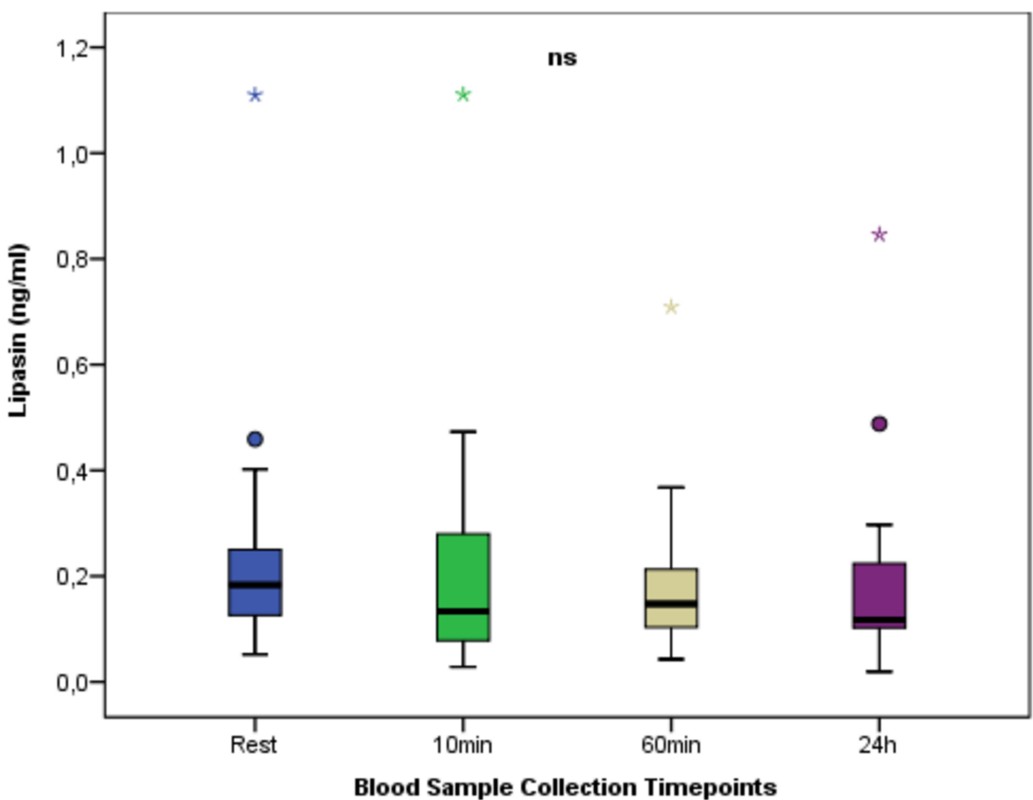

**Figure 4  Serum Lipasin concentrations at rest and after anaerobic performance.** The change in lipasin between times. Regarding lipasin values at different time points, no statistically significant difference was found between the measurement times ($p > 0.05$). ns: non-significant $p > 0.05$; Mean ± Sd; Rest:0.25 ± 0.24; 10 min:0.22 ± 0.25; 60min:0.19 ± 0.15; 24 h:0.20 ± 0.19; $p = 0.722$.

*al., 2013*). Similar observations were reported in an animal study in which both chronic and acute exercise increased apelin serum levels in healthy rats (*Ashraf & Roshan, 2012*; *Zarrinkalam & Heidarianpour, 2015*). A study on the effects of aerobic and resistance exercise on circulating apelin-12 and apelin-36 concentrations in middle-aged obese women indicated that both types of exercise significantly reduced apelin concentrations (*Jang et al., 2019*). In contrast, another human study reported that after 8 weeks of aerobic training in obese women, a decrease in plasma apelin levels was observed when body mass index and body fat percentage decreased simultaneously (*Sheibani, Hanachi & Refahiat, 2012*). In another study enrolled long-term resistance exercises, apelin levels remained unchanged (*Öner, 2019*). *Besse-Patin et al. (2014)* reported that there were no major changes in plasma apelin concentrations after exercise training in obese non-diabetic patients. Considering the findings of the abovementioned studies it can be said that the apelin response to acute exercise may be affected by the type of exercise. In the present study, no statistically significant difference was found between the apelin values measured at different time points (10 min, 60 min and 24 h later) after acute anaerobic exercise.

Lipasin is predominantly located within the cellular lining of the small blood vessels that surround skeletal muscles and within adipose tissue. An enzyme called LPL plays an

important role by breaking down fat molecules called triglycerides that are transported into the blood. When lipoprotein lipase breaks down triglycerides, the fat molecules are used by the body as energy or stored in fat tissue for later use (MedlinePlus Genetics). While LPL may indirectly affect energy metabolism, there is no direct evidence that it affects acute anaerobic performance. Anaerobic performance is primarily influenced by factors such as muscle strength, power, and endurance (*Franczyk et al., 2023*). Studies have shown that acute exercise can increase lipoprotein lipase activity (*Kantor et al., 1984*; *Yu et al., 1999*). In studies examining the changes in lipasin with exercise, it was reported that while lipasin levels were higher in obese individuals at the beginning, they decreased after the exercise program (*Abu-Farha et al., 2016*; *Keskin, 2019*).

In two different studies, it was noted that a single resistance exercise increased LPL activity in muscles (*Seip, Angelopoulos & Semenkovich, 1995*; *Greiwe, Holloszy & Semenkovich, 2000*). It was also indicated that LPL activity increased significantly within 4 h after treadmill walking (low intensity) exercise (*Bey & Hamilton, 2003*). *Hu et al. (2019)* reported that lipasin levels of the exercise group decreased significantly compared to pre-exercise levels in their study in which they utilized aerobic exercise 5 days a week and 30 min a day for 6 months in individuals with type 2 diabetes and obesity. Moreover, *Karaman, Arslan & Gürsu (2022)* examined the serum lipasin levels of four different groups (G1: control group; G2: metabolic syndrome control group; G3: metabolic syndrome + aerobic exercise; G4: metabolic syndrome + anaerobic exercise) in their study in which they applied six-week exercises on rats. Researchers showed that G2 and G3 lipasin levels were significantly higher than G1 lipasin levels, there was no significant difference between G3 and G4 groups, and G4 had the lowest lipasin levels. Together, they suggested that the 6-week exercise did not change the lipasin values of the anaerobic exercise group compared to the control groups.

Aerobic exercise has been shown to affect blood lipid metabolism as it raises high density lipoprotein cholesterol (HDL-C) by increasing LPL concentration and activity in skeletal muscles (*Harrison et al., 2012*). In addition other studies have revealed that aerobic exercise increases LPL, an enzyme vital for HDL formation (*Rowland et al., 1996*; *Hui et al., 2015*). These findings suggest that different types of exercise, especially aerobic exercise, may have a positive effect on LPL activity and contribute to beneficial changes in lipid metabolism and cardiovascular health. It is plausible that there is no difference in lipasin in the present study, given that the primary energy resource in anaerobic performance is carbohydrates.

## LIMITATIONS

This study is not without limitations. Ourstudy was conducted with a relatively small sample size. The research focused only on football and male football players. Therefore, the results may not be valid for female athletes and athletes in other sports. The study primarily investigated the immediate effects of acute anaerobic exercise on adipokine levels. Long-term effects or repeated periods of exercise have not been investigated. Hence, further studies are required to provide additional insight into the relationship between exercise and adipokine regulation.

## CONCLUSION

This study showed that ZAG concentration increased after acute anaerobic exercise. The results suggest that anaerobic exercise may be a potent stimulus for increasing circulating ZAG concentration. It was concluded that apelin and lipasin were not affected by acute anaerobic exercise.

## ACKNOWLEDGEMENTS

We would like to thank all the athletes, coaching staff, and managers for their support during the study measurements.

### Funding

This study was supported by Ondokuz Mayıs University's Scientific Research and Development Support Program Project, project number: PYO.YDS.1902.22.001. The funders had no role in study design, data collection and analysis, decision to publish, or preparation of the manuscript.

### Grant Disclosures

The following grant information was disclosed by the authors:
Ondokuz Mayıs University's Scientific Research and Development Support Program Project: PYO.YDS.1902.22.001.

### Competing Interests

The authors declare there are no competing interests.

### Author Contributions

- Şaban Ünver conceived and designed the experiments, performed the experiments, analyzed the data, prepared figures and/or tables, authored or reviewed drafts of the article, and approved the final draft.
- İlknur Bıyık conceived and designed the experiments, performed the experiments, authored or reviewed drafts of the article, and approved the final draft.
- Tülin Akman conceived and designed the experiments, performed the experiments, analyzed the data, authored or reviewed drafts of the article, and approved the final draft.
- Emre Şimşek conceived and designed the experiments, performed the experiments, analyzed the data, prepared figures and/or tables, authored or reviewed drafts of the article, and approved the final draft.
- Hamza Küçük conceived and designed the experiments, performed the experiments, analyzed the data, prepared figures and/or tables, authored or reviewed drafts of the article, and approved the final draft.
- Abdurrahim Kaplan conceived and designed the experiments, performed the experiments, analyzed the data, prepared figures and/or tables, authored or reviewed drafts of the article, and approved the final draft.

- Deniz Günay Derebaşı conceived and designed the experiments, performed the experiments, analyzed the data, prepared figures and/or tables, authored or reviewed drafts of the article, and approved the final draft.
- Selma İşler conceived and designed the experiments, performed the experiments, analyzed the data, prepared figures and/or tables, authored or reviewed drafts of the article, and approved the final draft.
- Canberk Çınar conceived and designed the experiments, performed the experiments, authored or reviewed drafts of the article, and approved the final draft.
- Tuba Kızılet conceived and designed the experiments, performed the experiments, authored or reviewed drafts of the article, and approved the final draft.
- Yeliz Tanrıverdi Çaycı conceived and designed the experiments, performed the experiments, authored or reviewed drafts of the article, and approved the final draft.

## Human Ethics

The following information was supplied relating to ethical approvals (*i.e.*, approving body and any reference numbers):

The study was approved by the 2020/46 ethics committee decision of the Ondokuz Mayıs University Clinical Research Ethics Committee. Moreover, this study was conducted in accordance with the ethical guidelines for human research of the Declaration of Helsinki.

## Data Availability

The data is available in the Supplemental File.

## Supplemental Information

Supplemental information for this article can be found online at http://dx.doi.org/10.7717/peerj.18093#supplemental-information.

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
