# Peer review of "Effect of acute anaerobic performance on zinc alpha 2 glycoprotein, apelin and lipasin levels"

_PeerJ, doi:10.7717/peerj.18093_

## Round 0.1 · original submission · Major Revisions

Sorry for the delay in getting this through the review system.

As you will see, both the reviewers are positive, but both have asked you some specific questions and seek clarification on certain points. While these do not, I think, require further experimental work, they will need a careful re-writing of the manuscript, particularly regarding its emphasis and generality to other studies (reviewer-2).

When you respond to these points, please also make corresponding changes in the text, even if you are rebutting the points raised. The fact that they have been raised here means others will have similar concerns/questions, so even if you rebut the central tenant of the comment, please address it in the final paper.

I hope these comments are of use, and look forward to seeing the revised paper.

·

Basic reporting

Language/Spelling

38: remove comma at either ‘Acute, or sudden,’
41: whilst instead of ‘while’
42: have instead of ‘has’
43: have instead of ‘has’
60: remove comma after ‘vasculature’
117: Can close brackets after ‘(at rest’
132: Figure 3 needs to be changed to ‘Figure 2’
134: Change comma in p value to period
136: ‘after exercise’ can be removed

138-139: Apelin has a lowercase ‘a’ on line 138 and an uppercase ‘A’ on 139. Either or for consistency
140: Change comma in p value to period
144: Change comma in p value to period
192: After lipoprotein lipase, include the acronym (LPL) or at line 68 as this is when you first mention it.
215: Include full text for HDL -high density lipoprotein- and then the acronym in brackets.
216: If added earlier, the ‘lipoprotein lipase’ can be emitted
217: can us LPL instead of lipoprotein lipase


Figures
Figure 1 Study Design
Step 2: Anaerobic has a spelling mistake.
Step 3: There is a period between ‘60th’ and ‘minute’
Figure is eligible but blurry.

Figure 2
**p value has a comma instead of a period
Graph could show all 18 data points in box plots (All graphs)
X axis title could be changed to ‘Blood Sample Collection Timepoints’

Figure 3
Where has the negative value come from in the ‘Rest’ group? No negative value in the raw data.

Figure 4
Same here with the negative values.
Title: Lipasin has no capital if wanting to be consistent with ZAG and Apelin in the other figure titles.

Raw data
No measurement parameters on headings. For example, weight (kg) height (m) ZAG (ng/ml).
Otherwise data is clear and can be opened.

Experimental design

• Are athletes monitored for the 24 hour period after the exercise/experiment? Effort has been made to highlight that athletes do no training 2 days prior, so I was just wondering if the period after was also regimented? For example, exercise free after the sprints, same cool-down routines with no ice baths and diet.

• Is this study blind?

Validity of the findings

• Great deal of variation in basal levels of adipokines ZAG, Apelin and Lipasin between athletes. With the sprint and blood collection only performed n=1, is this more aligned to the changes in individuals after exercise as apposed to the effect of exercise? Would need further repeats with the same athletes and same conditions.

• Unsure in what way the study supports ZAG being involved in energy and lipid metabolism? (Line 164) Good to show that it is elevated when compared to weight loss or long term exercise showing anaerobic respiration is affecting this.
With ZAG increasing after exercise and it's known contribution to lipid metabolism, from the blood samples can you measure glucose, triglycerides and FFA levels? This way there could be some metabolic data to pair with these ZAG results and support the energy and lipid metabolism statement.

Additional comments

Line 76-79 mention exercises effect on the immune system. The rest of the paper discusses adipokines and metabolism, so I just felt this small section was disjointed from the rest of the material and didn't contribute to the overall narrative.

Reviewer 2 ·

Basic reporting

The writing could be improved.

Experimental design

The description of methods needs some attention.

Validity of the findings

The statistics and conclusions could be improved to make better statements.

Additional comments

Introduction

I recommend reviewing the second paragraph (lines 47 to 56). You presented ZAG in the first and second lines, briefly discussed its function, and then repeated, “ZAG, a novel soluble adipokine with a molecular weight of approximately 43 kDa, has beneficial effects on energy metabolism and insulin sensitivity.” Additionally, you started every sentence in the paragraph with "ZAG."

Methods

What criteria did you use to classify the sample as athletes? You mentioned that players train four days a week. Are these players competing in any league (regional, national, or international)? How much training time do the athletes have?

In line 94, you mentioned “serious injury” but did not define it. Please provide a definition for “serious injury.”

In line 95, you did not include any chronic injuries and diseases. This criterion is very broad. Specify which types of injuries and diseases were not included in your study.

Did you perform the Kolmogorov-Smirnov test for each variable? I suggest verifying the Q-Q plots for your variables. In my analysis, your data do not fit a normal distribution.

Results

In line 132, I believe you made a mistake by referring to “Figure 3.”

I suggest including the p-values for each test.

Discussion

In lines 147 and 148, you wrote, “However, there was no significant change in apelin and lipasin concentrations.” This statement is incorrect. If you performed a power calculation, you could say, “We did not find (or detect) significant differences.” However, since you did not present any power calculation, you cannot make any judgment about "no significant values."

In line 164, you wrote, “The results obtained in our study support the role of ZAG in energy and lipid metabolism,” but I could not find this result in your study.

In lines 189 and 190, the phrase does not add any information. If you did not find a statistical difference, there is no reason to mention a “numeric increase.”

Conclusion

In lines 224 and 225, if you cannot generalize beyond your sample, what is the purpose of your study? How much did this limitation affect your study?

In lines 227 to 229, you wrote, “The study primarily examined the immediate effects of acute anaerobic exercise on adipokine levels. Long-term effects or repeated periods of exercise have not been investigated.” However, I do not think this is a limitation. Your purpose was to assess short-term effects only.

---

## Round 0.2 · Minor Revisions

As you will see, a few minor points remain to be clarified. Please ensure you address these, including details of the power calculation methodology.

Reviewer 2 ·

Basic reporting

no comment

Experimental design

The experimental design has been improved, but it still needs more details.

Validity of the findings

no comments

Additional comments

Thank you for addressing my questions, but I believe you missed some of them.

You did not define the level of your sample when referring to them as 'athletes.'

You did not respond to my request to clarify the definition of 'serious injuries.'

In line 150, you continue to state that 'there was no significant change in apelin and lipasin concentrations.' Please review my previous request regarding this.

You mentioned, 'as a result of the power analysis performed to generalize the results, it was determined that the data obtained from 18 athletes would be sufficient and generalizable.' However, I could not find any power analysis in your study.

---

## Round 0.3 · accepted · Accept

Thank you these final amendments. I am happy to accept this now.